# Influence of MRI Follow-Up on Treatment Decisions during Standard Concomitant and Adjuvant Chemotherapy in Patients with Glioblastoma: Is Less More?

**DOI:** 10.3390/cancers15204973

**Published:** 2023-10-13

**Authors:** Bart R. J. van Dijken, Annerieke R. Doff, Roelien H. Enting, Peter Jan van Laar, Hanne-Rinck Jeltema, Rudi A. J. O. Dierckx, Anouk van der Hoorn

**Affiliations:** 1Department of Radiology, Medical Imaging Center, University Medical Center Groningen, University of Groningen, 9700 RB Groningen, The Netherlands; 2Department of Neurology, University Medical Center Groningen, University of Groningen, 9700 RB Groningen, The Netherlands; r.h.enting@umcg.nl; 3Department of Radiology, Hospital Group Twente, 7600 SZ Almelo, The Netherlands; 4Department of Neurosurgery, University Medical Center Groningen, University of Groningen, 9700 RB Groningen, The Netherlands; j.r.jeltema@umcg.nl; 5Department of Nuclear Medicine, Medical Imaging Center, University Medical Center Groningen, University of Groningen, 9700 RB Groningen, The Netherlands

**Keywords:** glioblastoma, magnetic resonance imaging, treatment response assessment, treatment follow-up, pseudoprogression, perfusion imaging

## Abstract

**Simple Summary:**

Glioblastomas are brain tumors with a poor prognosis, and early tumor progression occurs often. Therefore, patients are closely monitored with regular MRI scans, usually at 2–3 month intervals. However, there is no evidence for this strategy, and it is not known if patients benefit from this approach. Furthermore, effects from the treatment sometimes mimic tumor progression (pseudoprogression). Pseudoprogession can cause uncertainty and makes decision making about continuing or stopping treatment difficult. This study evaluated how often standard scheduled MRI scans influenced treatment decisions and how often MRI scans caused uncertainty. Standard scheduled follow-up MRI scans rarely led to treatment consequences (<10%). However, many MRI scans caused diagnostic uncertainty (>25%). When scans were made at unscheduled timepoints, e.g., in patients with new or worsening symptoms, they had more consequences. Our results do not support the current pragmatic follow-up strategy and suggest a more tailored follow-up approach for glioblastoma patients.

**Abstract:**

MRI is the gold standard for treatment response assessments for glioblastoma. However, there is no consensus regarding the optimal interval for MRI follow-up during standard treatment. Moreover, a reliable assessment of treatment response is hindered by the occurrence of pseudoprogression. It is unknown if a radiological follow-up strategy at 2–3 month intervals actually benefits patients and how it influences clinical decision making about the continuation or discontinuation of treatment. This study assessed the consequences of scheduled follow-up scans post-chemoradiotherapy (post-CCRT), after three cycles of adjuvant chemotherapy [TMZ3/6], and after the completion of treatment [TMZ6/6]), and of unscheduled scans on treatment decisions during standard concomitant and adjuvant treatment in glioblastoma patients. Additionally, we evaluated how often follow-up scans resulted in diagnostic uncertainty (tumor progression versus pseudoprogression), and whether perfusion MRI improved clinical decision making. Scheduled follow-up scans during standard treatment in glioblastoma patients rarely resulted in an early termination of treatment (2.3% post-CCRT, 3.2% TMZ3/6, and 7.8% TMZ6/6), but introduced diagnostic uncertainty in 27.7% of cases. Unscheduled scans resulted in more major treatment consequences (30%; *p* < 0.001). Perfusion MRI caused less diagnostic uncertainty (*p* = 0.021) but did not influence treatment consequences (*p* = 0.871). This study does not support the current pragmatic follow-up strategy and suggests a more tailored follow-up approach.

## 1. Introduction

Glioblastomas are the most frequently occurring malignant brain tumors and have a poor prognosis with a median survival of only 15 months despite treatment [1]. Disease progression and tumor recurrence in glioblastoma are unavoidable and contribute to this dismal prognosis. In approximately half of glioblastoma patients, progression and recurrence already occur during the course of treatment [1,2]. Standard treatment in glioblastoma consists of maximal safe neurosurgical resection, followed by 30 daily fractions of 2 Gy radiotherapy to a total of 60 Gy with concomitant chemotherapy and subsequently six 28-day cycles of adjuvant chemotherapy with temozolomide, the so-called Stupp regimen [1]. A regular radiological follow-up is performed for the treatment response assessment and timely detection of tumor progression, and it is essential for clinical decision making about the continuation or discontinuation of treatment.

Multisequence magnetic resonance imaging (MRI) is the gold standard for radiological follow-up and is regularly acquired during the different phases of standard treatment and thereafter [3,4]. The reliable assessment of radiological follow-up, however, is limited by the occurrence of treatment effects mimicking tumor progression, such as pseudoprogression [5,6,7,8]. Therefore, decisions about the discontinuation of treatment are often postponed in asymptomatic patients with a new or increased contrast enhancing lesion on follow-up MRI scans less than three months after the end of radiotherapy [9]. Advanced sequences such as perfusion MRI are increasingly employed in clinical practice as they contribute to a better differentiation between tumor progression and treatment effects [6].

Currently, there is no consensus regarding the optimal interval for radiological follow-up during treatment [10]. The interval of scheduled MRI follow-up scans therefore differs between centers and countries. However, most centers perform an MRI at the end of the concomitant chemoradiation phase and after every two to three cycles of adjuvant chemotherapy [2,4,10]. It is unknown if this pragmatic strategy actually benefits the patient and how it influences clinical decision making about the continuation or discontinuation of treatment. Moreover, it is uncertain how much perfusion MRI contributes to clinical decision making when suspected progression occurs during treatment.

The aim of this study was (1) to assess the consequences of scheduled and unscheduled follow-up scans on treatment decisions during standard concomitant and adjuvant treatment in patients with glioblastoma. Since treatment decisions are often postponed due to the occurrence of treatment effects, we hypothesized that the impact of the scheduled scans during treatment is relatively low, and thus questioned whether all scheduled scans are necessary. In addition, (2) we evaluated how often follow-up scans resulted in diagnostic uncertainty, such as the inability to differentiate tumor progression from pseudoprogression, and (3) whether the application of perfusion MRI improved clinical decision making and led to less diagnostic uncertainty.

## 2. Materials and Methods

Adult patients with a histopathologically confirmed glioblastoma, who received standard treatment according to the Stupp protocol [1] between 2004 and 2019 at our tertiary university hospital, were retrospectively included if they underwent at least one follow-up MRI scan during treatment. Exclusion criteria were other intracranial lesions, previous radiotherapy to the brain, and a history of previous neurosurgery other than for the glioblastoma. Patients receiving chemotherapy regimens other than temozolomide were also excluded. This retrospective cohort study was approved by the local ethics committee and the need for written informed consent was waived. None of the included patients objected against the use of their anonymized data for research purposes.

### 2.1. Data Extraction

Clinical data were extracted from the electronic patient records. Patients underwent maximal safe neurosurgical resection or biopsy to confirm diagnosis. Resected or biopsied tissue was evaluated by a neuropathologist and, in more recent years, molecular markers were determined, including the methylation of O6-methylguanine-DNA methyl-transferase (MGMT) gene and the isocitrate dehydrogenase (IDH) mutation status.

Scan data consisted of scheduled follow-up scans and unscheduled scans. Scheduled follow-up scans were acquired at four standard timepoints: (1) <72 h post-operatively (post-OP); (2) four weeks after the completion of concomitant chemoradiotherapy (post-CCRT); (3) after three of six cycles of adjuvant temozolomide chemotherapy (TMZ3/6); and (4) after the completion of six cycles of adjuvant temozolomide chemotherapy (TMZ6/6). Unscheduled scans were divided into two categories: (I) scans that were acquired due to new or worsened clinical symptoms such as headache, nausea, vomiting, or neurological deficits; and (II) extra scans due to uncertainty on the previous (scheduled) scan.

For each scan, its consequence on the ongoing treatment was assessed. Decisions from the multidisciplinary tumor board meetings were assessed. Treatment consequences were treated as categorized variables and divided into three groups: (a) no treatment consequences, thus treatment was continued as scheduled; (b) minor consequences, which were defined as treatment adjustments without influencing the Stupp protocol, such as initiating or changing the dosage of corticosteroids or anti-epileptics, but continuing adjuvant chemotherapy; and (c) major treatment consequences that led to the interruption or early termination of the Stupp protocol, second-line chemotherapy including experimental regimens, or neurosurgical re-resection. In addition to the treatment consequences, it was assessed whether the scan led to diagnostic uncertainty, meaning it was unclear whether the imaging changes were due to tumor progression or treatment effects (pseudoprogression). The post-OP scans were left out of this analysis since pseudoprogression was not expected to occur this early in the course of the disease.

### 2.2. MRI Acquisition

MRI images were acquired on either a 1.5T or a 3.0T scanner. Various imaging protocols were followed, which always included anatomical sequences (pre- and post-contrast T1, T2, and fluid attenuated inversion-recovery (FLAIR)) with diffusion weighted imaging (DWI) or diffusion tensor imaging (DTI), and with or without perfusion weighted imaging (PWI). For the anatomical sequences, a pre- and post-contrast 3D T1-weighted MPRage (Repetition time [TR] 2100–2300 ms, echo time [TE] 2.32–2.67 ms, inversion time [TI] 900 ms, flip angle 8 degrees, slice thickness 1 mm, voxel size 1.0 × 1.0 × 1.0 mm^3^), a transverse T2-weighted turbo spin echo (TSE) sequence (TR 4630–8800 ms, TE 92–100 ms, flip angle 150 degrees, slice thickness 3–5 mm, voxel size 0.4 × 0.4 × 3.0–5.0 mm^3^), and a 3D (FLAIR) (TR 5000 ms, TE 337–391 ms, TI 1800 ms, slice thickness 1 mm, voxel size 1.0 × 1.0 × 1.0 mm^3^) were acquired. For DWI, a transverse RESOLVE sequence was acquired (TR 4440 ms, TE 60–104 ms, flip angle 180 degrees, slice thickness 4 mm, voxel size 1.1 × 1.1 × 4.0 mm^3^) with two b-values of 0 and 1000 s/mm^2^. In cases where DTI was used instead of DWI, a transverse DTI was acquired with an echo-planar imaging (EPI) sequence (TR 5300 ms, TE 93 ms, slice thickness 5 mm, voxel size 0.7 × 0.7 × 5.0 mm^3^) with 12 diffusion directions using b-values of 0 and 1000 s/mm^2^. For PWI, a transverse dynamic susceptibility contrast (DSC) echo-planar imaging (EPI) sequence was used (TR 1780 ms, TE 30 ms, flip angle 90 degrees, slice thickness 4 mm, voxel size 0.9 × 0.9 × 4.0 mm^3^) which was acquired during the administration of a gadolinium-based contrast agent (Dotarem) at an infusion rate of 4 mL/s. A pre-bolus of ¼ dose was used before the acquisition of PWI.

### 2.3. Statistical Analysis

All data were analyzed in SPSS version 23.0. Chi-square and Fisher’s exact tests were used to assess differences between categorical variables. Generalized estimating equations (GEE) were used since there were repeated measurements and to account for missing and clustered data. A two-sided *p*-value < 0.05 was used throughout this study.

## 3. Results

There were 261 patients included in this study with a median age of 59 years and of whom 164 (62.8%) were men. A majority of 151 patients (57.9%) completed standard treatment, whilst in the remaining 110 patients (42.1%) treatment was terminated early. Median survival was 15 months. Table 1 provides an overview of the general characteristics of the patients included.

### 3.1. Treatment Consequences

There were 790 scheduled scans acquired (Table 2): (1) post-OP in 188 patients (72%), (2) post-CCRT in 258 patients (98.9%), (3) TMZ3/6 in 190 patients (72.8%), and (4) TMZ6/6 in 154 patients (59%). For the post-OP timepoint, none of the 188 scans had any treatment consequences. This was also the case for 224 (86.8%) post-CCRT scans, 163 (85.8%) TMZ3/6 scans, and 130 (84.4%) TMZ6/6 scans. Minor treatment consequences occurred after eleven (4.3%) post-CCRT scans, eight (4.2%) TMZ3/6 scans, and three (1.9%) TMZ6/6 scans. There were 23 (8.9%), 19 (10%), and 21 (13.6%) major treatment consequences after the post-CCRT, TMZ3/6, and TMZ6/6 scans, respectively (Figure 1A). In the majority of these cases, other variables such as clinical symptoms or a poor clinical condition of the patient also contributed to the treatment decision of the multidisciplinary tumor board. When regarding the cases in which the decision was based on the MRI alone, major consequences occurred in six patients (2.3%) after the post-CCRT scan, six patients (3.2%) after the TMZ3/6 scan, and twelve patients (7.8%) after the TMZ6/6 scan (Figure 1B).

Among the included patients, 56 unscheduled scans were performed during treatment. Unscheduled scans were most commonly acquired during the adjuvant chemotherapy phase with a minority being acquired during concomitant chemoradiotherapy. The reasons for these unscheduled scans were (I) new or worsened clinical symptoms in 25 cases (44.6%) and (II) uncertainty on the previous (scheduled) scan in 31 cases (55.4%). In the 25 cases of new or worsened clinical symptoms, 15 scans (60%) did not lead to any treatment consequences. Two scans (8%) had minor consequences and eight scans (32%) led to major consequences. For the 31 scans that were acquired due to uncertainty on the previous MRI, 21 scans (67.8%) had no consequences, whilst 1 (3.2%) and 9 (29%) scans resulted in minor and major treatment consequences, respectively.

Comparing the scheduled scans (post-CCRT, TMZ3/6, and TMZ6/6) and unscheduled scans, major treatment consequences occurred after 62/602 (10.5%) scheduled scans (50% early termination, 15% re-resection or re-irradiation, 35% experimental second line regimen) and after 17/56 (30.4%) unscheduled scans (65% early termination, 23% re-resection, 12% experimental second line regimen). This difference (Figure 2) was statistically significant (*p* < 0.001).

### 3.2. Diagnostic Uncertainty on MRI

During the course of standard treatment, new contrast enhancement or a significant increase in an already present contrast-enhancing lesion was seen in 175 patients (67%). For the 602 scheduled scans, 167 (27.7%) resulted in diagnostic uncertainty about the effects of treatment. Diagnostic uncertainty occurred in 11 of the 56 (19.6%) unscheduled scans. There was no statistical difference between scheduled and unscheduled scans (*p* = 0.192). For the post-CCRT, TMZ3/6, and TMZ6/6 timepoints, 88 (34.1%), 55 (28.9%), and 24 (15.6%) scans caused diagnostic uncertainty, respectively. The post-CCRT (*p* = 0.003) and TMZ3/6 (*p* < 0.001) timepoints both differed significantly from the TMZ6/6 timepoint. There was no significant difference between the post-CCRT and TMZ3/6 timepoints (*p* = 0.247). For the unscheduled scans, three (12%) of the scans acquired due to new or worsened clinical symptoms resulted in diagnostic uncertainty, whilst this was the case for eight (25.8%) scans that were acquired due to uncertainty on the previous MRI (*p* = 0.321). Eventually, it was established by the multidisciplinary decision in the tumor board that the new or increased contrast enhancement was due to true tumor progression in 74 patients (42.3%) and pseudoprogression in 48 patients (27.4%) based on the RANO criteria [9]. For the remaining 53 patients (30%), it was not established whether the imaging changes were due to tumor progression or treatment effects by the end of standard adjuvant treatment.

### 3.3. Perfusion Weighted Imaging

Perfusion weighted imaging was acquired in 56% of the scheduled follow-up MRI scans (post-CCRT, TMZ3/6, and TMZ6/6). The inclusion of a perfusion sequence in the imaging protocol did not result in more major treatment consequences compared to protocols without perfusion imaging (*p* = 0.871). However, there was less diagnostic uncertainty following scheduled scans that incorporated a perfusion sequence (56%) compared to scheduled scans without perfusion imaging (44%) (*p* = 0.021).

## 4. Discussion

This retrospective longitudinal study demonstrated that approximately 90% of the scheduled follow-up scans during standard concomitant and adjuvant treatment in 261 glioblastoma patients did not result in any treatment consequences, whilst diagnostic uncertainty was caused by about a quarter of all acquired scans. Furthermore, unscheduled scans led to significantly more treatment consequences than the scheduled scans. The incorporation of perfusion MRI in the imaging protocol resulted in less diagnostic uncertainty without an impact on the treatment. Our results do not support the current radiological follow-up schemes during standard treatment in glioblastoma patients, and suggest that at least one of the scheduled scans could potentially be omitted.

The current practice of scheduled MRI follow-up scans at predetermined intervals during standard treatment and scheduled routine surveillance after the completion of standard treatment is not evidence-based [10]. A previous study by Monroe et al. investigated how often tumor progression was discovered on scheduled follow-up scans versus unscheduled scans due to the development of symptoms [4]. The authors found that 63.5% of the patients with tumor progression were detected with scheduled surveillance imaging [4]. However, half of the patients in the surveillance detection group were also experiencing symptoms at the time of the scheduled scan [4]. Another recent study aimed to establish the optimal follow-up interval of glioblastoma patients by using parametric modeling of standardized progression-free survival curves [2]. An interval of 7–8 weeks between scans for at least two years was found to be optimal [2]. However, this study only investigated the period after the completion of standard treatment, whereas our study focused on the radiological follow-up during concomitant and adjuvant treatment. The suggested interval of 7–8 weeks approaches the estimated volume doubling time of 49.6 days for untreated glioblastomas according to an in vivo MRI study [11]. A large survey among all neuro-oncology centers in the United Kingdom demonstrated that a lot of variation exists in predefined scan timepoints between centers [12]. Most centers (>70%) performed standard MRI follow-up scans early post-operative, during adjuvant treatment, and after the completion of standard adjuvant treatment, comparable to the scheme used in our hospital [12]. Currently, a large multicenter retrospective cohort study is undertaken (INTERVAL-GB) to assess MRI monitoring practice during and after standard treatment in the United Kingdom and Ireland, and how this influences survival [13].

Our study showed that only a low number (8.9–13.6%) of scheduled follow-up MRI scans at predetermined intervals led to major treatment consequences, whilst diagnostic uncertainty often occurred. Additionally, the incidence of early treatment termination was even lower when decisions about treatment continuation or discontinuation were made based on the radiological information alone (2.3–7.8%). Furthermore, unscheduled scans in symptomatic patients more often led to major treatment consequences than these predetermined scheduled scans in asymptomatic patients. This further emphasizes the problem with the current pragmatic approach and raises the question of whether one or more of these scheduled follow-up MRI scans during standard concomitant and adjuvant treatment could potentially be omitted or replaced by on-demand scans for new or worsening clinical symptoms.

Certain timepoints could still have clinical value despite a low number of major treatment consequences, however. None of the post-OP scans in our cohort led to any treatment consequences. This is in line with a previous study that demonstrated no difference in the number of patients that received chemoradiation therapy between glioblastoma patients with and without a post-operative scan [14]. The same study showed that the use of post-operative MRI did not result in a survival benefit for patients [14]. However, post-operative MRI is important in determining the extent of resection, which is an established important prognostic factor in glioblastoma [15,16,17]. Additionally, post-operative MRI can reveal possible surgical complications and is used as a reference image to interpret subsequent MRI scans. The post-CCRT timepoint serves as a baseline scan after radiotherapy and also visualizes potential radiotherapy related adverse effects [18,19]. The TMZ6/6 timepoint marks the end of standard adjuvant treatment and can thus be used as a baseline scan for experimental second-line therapeutic regimens. The TMZ6/6 scan also had the highest number of major treatment consequences. The TMZ3/6 scan, however, lacks a pragmatic rational such as the post-OP or post-CCRT timepoints.

Treatment-induced changes such as pseudoprogression causing diagnostic uncertainty occur frequently during treatment in glioblastoma patients [20]. For our cohort, diagnostic uncertainty occurred in 27.7% of scheduled follow-up scans, which is comparable to the literature [20]. Scheduled scans early in the treatment scheme, such as the post-CCRT and TMZ3/6, resulted in more diagnostic uncertainty than the TMZ6/6 scan at the end of standard treatment. It is known that diagnostic uncertainty can induce patient and caregiver anxiety [21,22,23]. Several studies among cancer patients have evaluated follow-up scan-associated distress, sometimes referred to as “scanxiety”, and found a negative impact on quality of life [24,25,26]. Furthermore, for glioblastoma patients specifically, the quality of life was found to be lowest during adjuvant therapy [27]. If the TMZ3/6 scan were to be omitted, this would result in 29% less diagnostic uncertainty based on our results. However, a “negative” MRI scan demonstrating no signs of radiological progression could also be reassuring and reduce anxiety and distress [28]. Currently, there are no data available on the influence of diagnostic uncertainty versus reassurance in glioblastoma patients during treatment, and future studies on this topic are warranted.

Perfusion MRI is becoming increasingly important in radiological follow-up protocols for glioblastoma, as perfusion MRI has a high diagnostic accuracy for differentiating tumor progression from treatment effects [6,29,30]. Indeed, the use of perfusion MRI led to less diagnostic uncertainty on follow-up scans in our cohort. However, perfusion MRI did not influence the consequences of the scan on treatment decisions. A possible explanation is that perfusion MRI has only become the standard of care in recent years, and the clinical experience and quantification of this technique have expanded since [31]. Another explanation could be that although perfusion has a high diagnostic accuracy of >85% [6], it is not 100%. Clinicians frequently give patients the benefit of the doubt and continue treatment as standard second line treatment is not available.

This study has several limitations. Firstly, its retrospective nature may have resulted in a sampling bias, since perfusion MRI and genetic biomarkers only became the standard of care in recent years. The number of patients for whom the IDH mutation and MGMT methylation status are known are hence relatively low. Secondly, the number of patients who were scanned on 3.0T MRI scanners was too low to allow for subanalysis. It is, however, unlikely that these mutations or differences in field strength would impact the results. Furthermore, this is a single-center study and may not be fully applicable to other centers or countries. A multicenter study is currently ongoing, and it will be interesting to see if its results are in line with our conclusions [12]. Finally, the end point of this study was the completion of standard treatment, and the subsequent follow-up period of active surveillance was therefore not analyzed.

## 5. Conclusions

Scheduled follow-up scans during standard treatment in glioblastoma patients rarely have major treatment consequences such as an early termination of treatment, but do introduce considerable diagnostic uncertainty. The use of perfusion MRI does not impact treatment decision making, but significantly reduces diagnostic uncertainty. This study does not support the current pragmatic approach with standard scheduled follow-up MRI scans. Potentially, one or more scheduled MRI scans could be omitted or replaced by unscheduled scans in symptomatic patients.

## Figures and Tables

**Figure 1 cancers-15-04973-f001:**
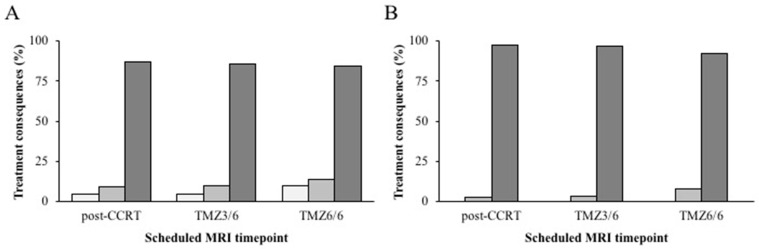
Treatment consequences for scheduled follow-up scans. Percentage of treatment consequences (*y* axis) for the different scheduled follow-up MRI scan timepoints (*x* axis). Treatment consequences based on MRI together with clinical parameters (**A**) and treatment consequences solely based on MRI (**B**) are shown. Number of minor (white bars), major (light gray bars), and no consequences (dark gray bars) on treatment for each timepoint are shown. The post-operative (post-OP) timepoint is not included as it never resulted in any consequences on treatment. Abbreviations of scan timepoints: post-CCRT = post-concomitant chemoradiotherapy; TMZ3/6 after three cycles of adjuvant temozolomide chemotherapy; TMZ6/6 = after six cycles of adjuvant temozolomide chemotherapy which marks the completion of standard treatment.

**Figure 2 cancers-15-04973-f002:**
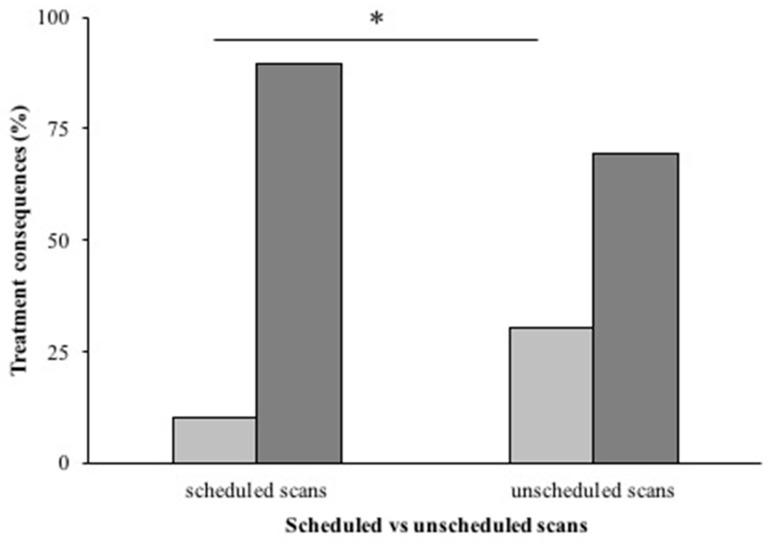
Treatment consequences for scheduled and unscheduled scans. Percentage of major treatment consequences (in light gray bars) compared to no or minor treatment consequences (dark gray bars) for both scheduled follow-up scans (left) and unscheduled scans (right). Unscheduled scans resulted in significantly more major treatment consequences than scheduled scans (* = *p* < 0.001).

**Table 1 cancers-15-04973-t001:** General characteristics.

Variable		Number	%
Number of patients		261	
Median age at diagnosis		59 years	
Median overall survival		15 months	
Sex	Men	164	62.8
	Women	97	37.2
IDH mutation	IDH-1	10	3.8
	Wild type	84	32.2
	Unknown	167	64.0
MGMT status	Methylated	13	5.0
	Unmethylated	8	3.1
	Unknown	240	92.0
Extent of resection	Gross-total	79	30.3
	Sub-total	120	46.0
	Biopsy	39	14.9
	Unknown	23	8.8
Completion of Stupp protocol	Yes	151	57.9
	No	110	42.1

Abbreviations: IDH = isocitrate dehydrogenase; MGMT = O6-methylguanine-DNA methyl-transferase.

**Table 2 cancers-15-04973-t002:** Number of scheduled scans and used sequences.

Timepoint		Field Strength		Total
		1.5T	3.0T	N
Post-OP	N (%)	159 (84.6)	29 (15.4)	188
Post-CCRT	N (%)	256 (99.2)	2 (0.8)	258
TMZ3/6	N (%)	188 (98.2)	2 (1.1)	190
TMZ6/6	N (%)	154 (100)	0 (0)	154
Total	N (%)	757 (95.8)	33 (4.2)	790

Scheduled scan time points: Post-OP = post-operative; post-CCRT = post concomitant chemoradiotherapy; TMZ3/6 = after three cycles of adjuvant temozolomide chemotherapy; TMZ6/6 = after six cycles of adjuvant temozolomide chemotherapy which marks the completion of standard treatment.

## Data Availability

The data presented in this study are available on request from the corresponding author.

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
