# Peer review of "Influence of MRI Follow-Up on Treatment Decisions during Standard Concomitant and Adjuvant Chemotherapy in Patients with Glioblastoma: Is Less More?"

_cancers, 2023, doi:10.3390/cancers15204973_

Round 1
Reviewer 1 Report
The article outlines the consequences of scheduled and unscheduled MRI in glioblastoma patients during the treatment and the potential use of perfusion MRI to reduce diagnostic uncertainty. The article is well written, structured and comprehensive.
Improvement to introduction by giving brief background on glioblastoma and its occurrence rate can be included.
The article evaluates the consequences of scheduled follow-up scans post-concomitant chemoradiotherapy, after 3 cycles of adjuvant chemotherapy with temozolomide [TMZ3/6], after completion of treatment [TMZ6/6] and unscheduled scans of glioblastoma patient cohort undergoing standard concomitant and adjuvant treatment. Additionally, they have evaluated how often follow-up scans resulted in diagnostic uncertainty and whether advanced MRI technique, namely perfusion MRI improved clinical decision making. The authors suggest the omission of one or more scheduled MRI scans for symptomatic patients and to replace it with unscheduled scans and to use perfusion MRI to reduce diagnostic uncertainty. The article can be accepted after the comments raised are addressed.
Comments:
1) There is no general agreement for the optimal interval of radiological follow-up for glioblastoma patients during treatment. To minimize exposure risks, health care system overloading, and general cost-effectiveness, re-evaluation of current practices, aided by past evidence and the right modeling and validation tools is necessary. This article sheds light into that area.
2) In the data extraction part, the authors have assessed the diagnostic uncertainty were due to tumor progression or treatment effects (pseudoprogression). But they have omitted the post-op scans for this analysis. Explain why it was omitted.
3) Commendable use of advanced imaging techniques to support the claim.
4) While comparing scheduled and unscheduled scans, the authors have shown that major treatment consequences occurred after 62/602 (10.5%) scheduled scans and after 17/56 (30.4%) unscheduled scans (Figure 2). Unscheduled scans resulted in significantly more major treatment consequences than scheduled scans (* = p<0.001). Give more detailed description of the suggested major treatment consequences.
5) The authors have reported that the unscheduled scans in symptomatic patients more often led to major treatment consequences than the pre-determined scheduled scans in asymptomatic patients thereby questioning whether one or more of these scheduled follow-up MRI scans during standard concomitant and adjuvant treatment could potentially be omitted. This can only be applied for asymptomatic patients. For better decision making, advanced MRI techniques must be used regardless of symptoms.
6) The authors also claim that the “scanxiety” induced by the MRI diagnostic uncertainty in patients and caregivers could be reduced by the omission of TMS3/6 scan. But there is not enough evidence to support the relation between diagnostic uncertainty and reassurance in glioblastoma patients.
7) The authors make a bold statement about the unlikely effect of the results of the study without having complete data on the number of patients with IDH mutation, MGMT methylation status and the impact of field strength differences.
8) References need to be consistent. Majority of the references provided is written as et al except a few.
Author Response
Comment 1:
The article outlines the consequences of scheduled and unscheduled MRI in glioblastoma patients during the treatment and the potential use of perfusion MRI to reduce diagnostic uncertainty. The article is well written, structured and comprehensive.
The article evaluates the consequences of scheduled follow-up scans post-concomitant chemoradiotherapy, after 3 cycles of adjuvant chemotherapy with temozolomide [TMZ3/6], after completion of treatment [TMZ6/6] and unscheduled scans of glioblastoma patient cohort undergoing standard concomitant and adjuvant treatment. Additionally, they have evaluated how often follow-up scans resulted in diagnostic uncertainty and whether advanced MRI technique, namely perfusion MRI improved clinical decision making. The authors suggest the omission of one or more scheduled MRI scans for symptomatic patients and to replace it with unscheduled scans and to use perfusion MRI to reduce diagnostic uncertainty. The article can be accepted after the comments raised are addressed.
Response:
We would like to thank the reviewer for the extensive review of our work and the helpful comments. These suggestions have strengthened the quality of our manuscript.
Comment 2:
Improvement to introduction by giving brief background on glioblastoma and its occurrence rate can be included.
Response:
In line with the reviewer’s suggestion we have provided a sentence with more background information on glioblastoma epidemiology to the introduction section. We have added the following sentence to start the manuscript: “Glioblastomas are the most frequently occurring malignant brain tumors and have a poor prognosis with a median survival of only 15 months despite treatment [1].”
Comment 3:
There is no general agreement for the optimal interval of radiological follow-up for glioblastoma patients during treatment. To minimize exposure risks, health care system overloading, and general cost-effectiveness, re-evaluation of current practices, aided by past evidence and the right modeling and validation tools is necessary. This article sheds light into that area.
In the data extraction part, the authors have assessed the diagnostic uncertainty were due to tumor progression or treatment effects (pseudoprogression). But they have omitted the post-op scans for this analysis. Explain why it was omitted.
Response:
The reviewer correctly states that the assessment of the diagnostic uncertainty did not include the post-op time point. Our results showed that none of the 188 post-op scans had any treatment consequences, meaning the concomitant and adjuvant treatment was started in all 188 patients. Neither tumor growth, nor treatment effects – which are usually due to radiation and/or chemotherapy effects – are expected within 72 hours after surgery. Therefore, we did not include this timepoint for the diagnostic uncertainty assessment. To further clarify this, we have added the following sentence to the end of this paragraph: “(…) since pseudoprogression was not expected to occur this early in the course of the disease.”
Comment 4:
Commendable use of advanced imaging techniques to support the claim.
Response:
We thank the reviewer for this acknowledgment. Perfusion MRI is now available in many neuro-oncology centers and its potential in neuro-oncology is widely accepted due to the high diagnostic value. Indeed, we feel that the incorporation of perfusion MRI data has strengthened our manuscript.
Comment 5:
While comparing scheduled and unscheduled scans, the authors have shown that major treatment consequences occurred after 62/602 (10.5%) scheduled scans and after 17/56 (30.4%) unscheduled scans (Figure 2). Unscheduled scans resulted in significantly more major treatment consequences than scheduled scans (* = p<0.001). Give more detailed description of the suggested major treatment consequences.
Response:
We agree with the reviewer that it would be of interest to provide the major treatment differences for both scheduled and unscheduled scans in more detail. We have included the percentage of the different major treatment consequences for both groups. The sentence was changed accordingly: “Comparing the scheduled scans (post-CCRT, TMZ3/6 and TMZ6/6) and unscheduled scans, major treatment consequences occurred after 62/602 (10.5%) scheduled scans (50% early termination, 15% re-resection or re-irradiation, 35% experimental second line regimen) and after 17/56 (30.4%) unscheduled scans (65% early termination, 23% re-resection, 12% experimental second line regimen).”
Comment 6:
The authors have reported that the unscheduled scans in symptomatic patients more often led to major treatment consequences than the pre-determined scheduled scans in asymptomatic patients thereby questioning whether one or more of these scheduled follow-up MRI scans during standard concomitant and adjuvant treatment could potentially be omitted. This can only be applied for asymptomatic patients. For better decision making, advanced MRI techniques must be used regardless of symptoms.
Response:
We do agree with this more nuanced view of the reviewer. We argue that pre-determined scanning in asymptomatic patients during standard treatment might not have much value. However, according to our data, scanning symptomatic patients (with new or worsening clinical symptoms) remains pivotal. Therefore, we have added the following part to the final sentence of paragraph 3 of the discussion: This further emphasizes the problem about the current pragmatic approach and raises the question whether one or more of these scheduled follow-up MRI scans during standard concomitant and adjuvant treatment could potentially be omitted “or be replaced by on demand scans for new or worsening clinical symptoms.”
Comment 7:
The authors also claim that the “scanxiety” induced by the MRI diagnostic uncertainty in patients and caregivers could be reduced by the omission of TMS3/6 scan. But there is not enough evidence to support the relation between diagnostic uncertainty and reassurance in glioblastoma patients.
Response:
This is a very interesting issue. In cancer literature “scanxiety” is well described with a negative impact on quality of life. However, the reviewer is correct that for glioblastoma such evidence is lacking. We hypothesize that omitting the TMZ3/6 scan – since it has a low number of treatment consequences and lacks a pragmatic reason – could reduce diagnostic uncertainty for 55 of 190 patients in our cohort. But the reviewer is absolutely right that this remains hypothetical. Therefore, we have deleted this sentence from the manuscript.
Furthermore, the positive effects of a “negative” scan are not known. Hence, we have also incorporated this view in the paragraph and concluded the paragraph with the following sentence: “Currently, there is no data available on the influence of diagnostic uncertainty versus reassurance in glioblastoma patients during treatment and future studies on this topic are warranted.”
Comment 8:
The authors make a bold statement about the unlikely effect of the results of the study without having complete data on the number of patients with IDH mutation, MGMT methylation status and the impact of field strength differences.
Response:
The reviewer is correct. We did not have sufficient data for subanalyses on genetic profile (IDH and MGMT) or field strength differences. Therefore, we have discussed this as a limitation in the limitation section at the end of the discussion. We feel that it is very unlikely that these characteristics would have impacted the results, especially since both 1.5T and 3.0T systems are routinely used in clinical care (and many centers utilize both).
Comment 9:
References need to be consistent. Majority of the references provided is written as et al except a few.
Response:
In line with the reviewer’s comment and the journal’s guidelines we have changed the references. All authors up to 10 are now listed. Any references with >10 authors will have “et al” as per journal guidelines.
Reviewer 2 Report
I am delighted that I have the opportunity to review such an elegant paper presenting properly designed and conducted research based on clinical practice. The subject is absolutely up-to-date, the study group representative, and the conclusions are of interest.
I would like to make a comment not necessarily in line with the messege of the ms. What we now about GBM is that it grows fast (doubles its mass in few weeks) and will recurr (ca 6-9 mths in up to 90% of pts after surgery). Therefore the reason for standard schedule MRI is not to confirm the reason of problems when the clinical symptoms occur, but to detect the asymptomatic recurrence for additional treatment (mainly RTX now, also surgery, rarely CTX). Also the time intervals between the controls are the derivatives of tumor biology (doubling time). The authors don't have to share the view of course, but the second view should be better underlined in discussion.
The table summarizing the detailed treatment changes and timepoint of changes should be included (at least as supplement).
One thorough reading is recommended to detect smalll grammatical/spelling/punctuation/styllistic mistakes.
Author Response
Comment 1:
I am delighted that I have the opportunity to review such an elegant paper presenting properly designed and conducted research based on clinical practice. The subject is absolutely up-to-date, the study group representative, and the conclusions are of interest.
Response:
We thank this reviewer for the kind words and the comments provided below.
Comment 2:
I would like to make a comment not necessarily in line with the message of the ms. What we now about GBM is that it grows fast (doubles its mass in few weeks) and will recur (ca 6-9 mths in up to 90% of pts after surgery). Therefore, the reason for standard schedule MRI is not to confirm the reason of problems when the clinical symptoms occur, but to detect the asymptomatic recurrence for additional treatment (mainly RTX now, also surgery, rarely CTX). Also, the time intervals between the controls are the derivatives of tumor biology (doubling time). The authors don't have to share the view of course, but the second view should be better underlined in discussion.
Response:
The point raised by the reviewer is very interesting indeed. We have included data on the tumor volume doubling time, which is around 7 weeks, according to an in vivo MRI study (Stensjoen et al. Neuro Oncol 2015). Interestingly, another modeling study which we discussed in the discussion section also recommended a scan interval of 7-8 weeks for glioblastoma follow-up. We have added the following sentence: “The suggested interval of 7-8 weeks approaches the estimated volume doubling time of 49.6 days for untreated glioblastomas according to a in vivo MRI study [11].”
However, in our view it remains true that diagnostic uncertainty remains a challenge and that when scans do not prove to have any treatment consequences they might be omitted despite the theoretical value. However, the application of more advanced imaging techniques (such as perfusion MRI) could potentially overcome this hurdle.
Comment:
The table summarizing the detailed treatment changes and timepoint of changes should be included (at least as supplement).
Response:
We thank the reviewer for this comment and this is in line with comment 5 of reviewer #1 (“Give more detailed description of the suggested major treatment consequences”).
In response to this comment we have added details for the different major consequences of scheduled and unscheduled scans to the results section: “Comparing the scheduled scans (post-CCRT, TMZ3/6 and TMZ6/6) and unscheduled scans, major treatment consequences occurred after 62/602 (10.5%) scheduled scans (50% early termination, 15% re-resection or re-irradiation, 35% experimental second line regimen) and after 17/56 (30.4%) unscheduled scans (65% early termination, 23% re-resection, 12% experimental second line regimen).”